# CO-MOT: Boosting End-to-end Transformer-based Multi-Object Tracking via Coopetition Label Assignment and Shadow Sets

**Feng Yan**,[*] **Weixin Luo**,[*] **Yujie Zhong, Yiyang Gan, Lin Ma**[†]
Meituan Inc., China.
{yanfeng05, luoweixin, zhongyujie, ganyiyang}@meituan.com
forest.linma@gmail.com

## Abstract

Existing end-to-end Multi-Object Tracking (e2e-MOT) methods have not surpassed non-end-to-end tracking-by-detection methods. One possible reason lies in the training label assignment strategy that consistently binds the tracked objects with tracking queries and assigns few newborns to detection queries. Such an assignment, with one-to-one bipartite matching, yields an unbalanced training, *i.e.*, scarce positive samples for detection queries, especially for an enclosed scenew with the majority of the newborns at the beginning of videos. As such, e2e-MOT will incline to generate a tracking terminal without renewal or re-initialization, compared to other tracking-by-detection methods. To alleviate this problem, we propose Co-MOT, a simple yet effective method to facilitate e2e-MOT by a novel coopetition label assignment with a shadow concept. Specifically, we add tracked objects to the matching targets for detection queries when performing the label assignment for training the intermediate decoders. For query initialization, we expand each query by a set of shadow counterparts with limited disturbance to itself. With extensive ablation studies, Co-MOT achieves superior performances without extra costs, *e.g.*, 69.4% HOTA on DanceTrack and 52.8% TETA on BDD100K. Impressively, Co-MOT only requires 38% FLOPs of MOTRv2 with comparable performances, resulting in the $1.4\times$ faster inference speed. Source code is publicly available at https://github.com/BingfengYan/CO-MOT.

## 1 Introduction

Multi-Object Tracking (MOT) is traditionally tackled by a series of tasks, *e.g.*, object detection (Zhao et al., 2024; Zou et al., 2023), appearance Re-ID (Zheng et al., 2016; Li et al., 2018; Bertinetto et al., 2016; Ye et al., 2024), motion prediction (Lefèvre et al., 2014; Welch et al., 1995), and temporal association (Kuhn, 1955). The sparkling advantage of this paradigm is task decomposition, leading to an optimal solution for each task. However, it lacks global optimization for the whole pipeline.

Recently, there has been a rise in end-to-end Multi-Object Tracking (e2e-MOT) models using transformers. These models input consecutive video frames and directly output bounding boxes and association information, eliminating the need for pre- or post-processing steps such as separate detectors, Re-ID feature extraction, or IoU matching. Notable contributions in this field include MOTR (Zeng et al., 2022) and TrackFormer (Meinhardt et al., 2022), which perform detection and tracking simultaneously in unified transformer decoders. Specifically, tracking queries achieve identity tracking through recurrent attention over time. Meanwhile, detection queries discover newborns in each newly arriving frame, excluding previously tracked objects, due to a Tracking Aware Label Assignment (TALA) during training. However, the TALA matching mechanism often leads to an imbalance between detection queries and tracking queries. This mechanism first matches the tracking queries and then assigns the remaining ground truth objects (newborns) to the detection queries. In many scenarios, especially in closed environments, there are very few newborn objects in the video

---

[*]These authors contributed equally to this work.
[†]Corresponding author.

frames after the initial frame. To illustrate this, we conduct an analysis on the DanceTrack dataset and found that the ratio of newborn objects to tracked targets is 213:25483. Moreover, we observe that e2e-MOT tends to underperform due to suboptimal detection capabilities. In Figure 1, e2e-MOT consistently results in tracking termination. MOTRv2 (Zhang et al., 2023) supports this observation and addresses it by leveraging a pre-trained YOLOX detector to boost the performances. However, introducing an additional detector introduces extra overhead during deployment and undermines the advantages of the e2e-MOT approach.

In this paper, we present a novel perspective for addressing the above limitations of e2e-MOT: **detection queries are exclusive but also beneficial to tracking queries**. To this end, we develop a COopetition Label Assignment (COLA) for training tracking and detection queries. Except for the last Transformer decoder remaining the competition strategy to avoid trajectory redundancy, we allow the previously tracked objects to be reassigned to the detection queries in the intermediate decoders. Due to the self-attention mechanism among all queries, detection queries will be complementary to tracking queries with the same identity, resulting in feature augmentation for tracking objects with significant appearance variance. Thus, the tracking terminal problem will be alleviated.

Besides TALA, another drawback in transformer-based detection and tracking is one-to-one bipartite matching used, which cannot produce sufficient positive samples, as stated by Co-DETR (Zong et al., 2023) and HDETR (Jia et al., 2023) that introduce one-to-many assignment to overcome this limitation. Different from these remedies with one-to-many auxiliary training, we develop a **one-to-set matching strategy with a novel shadow concept**, where each individual query is augmented with multiple shadow queries by adding limited disturbance to itself, so as to ease the one-to-set optimization. The set of shadow queries endows CO-MOT with discriminative training by optimizing the most challenging query in the set with the maximal cost, which can thereby enhance the generalization ability.

We evaluate our proposed method on the public MOT benchmarks, including DanceTrack (Sun et al., 2022), BDD100K (Yu et al., 2020) and MOT17 (Milan et al., 2016), and achieve superior performances. The contributions of this work lie in threefold. i) We introduce a coopetition label assignment for training tracking and detection queries for e2e-MOT with high efficiency. ii) We develop a one-to-set matching strategy with a novel shadow concept to address the hunger for positive training samples and enhance generalization ability. iii) Our approach achieves superior performances on public benchmarks, while functioning as an efficient tool to boost the performance of end-to-end transformer-based MOT.

## 2 METHOD

### 2.1 MOTIVATION

To explore the shortcomings of current end-to-end methods in tracking, we conduct an in-depth study of the effectiveness on DanceTrack validation and MOT17 test dataset by analyzing MOTR, which is one of representative e2e-MOT methods. In Figure 1, we show the tracking results of MOTR in some video frames, *e.g.*, DanceTrack0073 and MOT17-09. In the left three columns of the first row, the 3rd person (in the yellow box) is tracked normally in frame #237. However, in frame #238, due to an inaccurate detection, the bounding box is not accurately placed around that person (the box is too large to include a person on the left side). In frame #239, the tracking is completely wrong and associated with the 2nd person instead. In the right three columns of the first row, the 2nd person (in the yellow box) is successfully detected and tracked in frame #302. However, in frame #312, this person is occluded by other people. When the person appears again in frame #322, she is not successfully tracked or even detected. To determine whether the tracking failure is caused by the detection or association of MOTR, we visualized MOTR's detection results in the second row. We remove the tracking queries during inference, and the visualization shows that all persons are accurately detected. This demonstrates that the detection will deteriorate due to the nearby tracked objects, though TALA used in training ensures that the detection with the same identity of tracked objects will be suppressed.

We further provide quantitative results of how the queries affect each other in Table 1. All the decoded boxes of both tracking and detection queries are treated as detection boxes, allowing evaluation by the mAP metric commonly used for object detection. We can see from the table that the

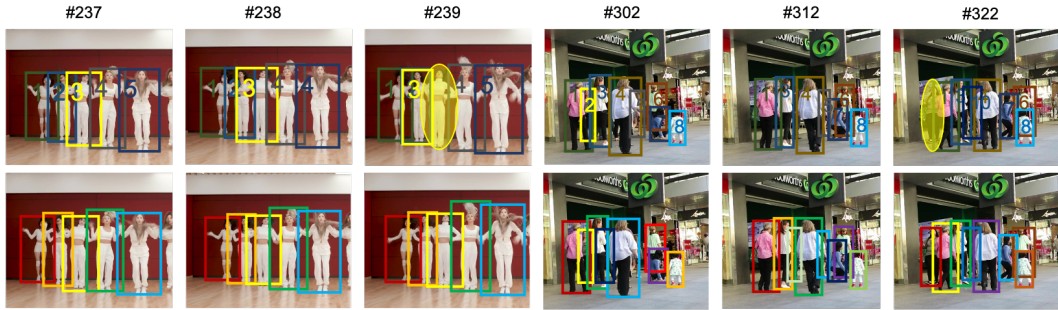

Figure 1: Visualization of tracking results in DanceTrack0073 and MOT17-09 videos. The first row displays the tracking results from MOTR, where all individuals are correctly initialized at the beginning (#237 and #302). However, heavy occlusion appears in the middle frames (#238 and #312), resulting in inaccurate detection (indicated by the yellow boxes). The tracking of the yellow targets finally terminates in frames #239 and #322. The second row shows MOTR's detection results, where tracking queries are removed during the inference process. Targets in different frames are accurately detected without interference from tracking queries.

Table 1: Detection performance (mAP) of MOTR (v2) on DanceTrack validation dataset. ✓means whether the tracking queries are used in the training or inference phase. All decoded boxes of both tracking if applicable and detection queries are treated as detection boxes for evaluation on mAP. We separately evaluate the detection performance for six decoders. For analysis, please refer to the motivation section.

|     | model | Training | Inference | 1 | 2 | 3 | 4 | 5 | 6 |
|-----|-------|----------|-----------|-----|-----|-----|-----|-----|-----|
| (a) | MOTR | ✓ | ✓ | 41.4 | 42.4 | 42.5 | 42.5 | 42.5 | 42.5 |
| (b) | MOTR | ✓ | | 56.8 | 60.1 | 60.5 | 60.5 | 60.6 | 60.6 |
| (c) | MOTR | | | 57.3 | 62.2 | 62.9 | 63.0 | 63.0 | 63.0 |
| (d) | MOTRv2 | ✓ | ✓ | 67.9 | 70.2 | 70.6 | 70.7 | 70.7 | 70.7 |
| (e) | MOTRv2 | ✓ | | 71.9 | 72.1 | 72.1 | 72.1 | 72.1 | 72.1 |
| (f) | CO-MOT(ours) | ✓ | ✓ | - | - | - | - | - | 69.1 |

vanilla MOTR (a) has a low mAP of 42.5%, but it increases by 18.1% (42.5% vs 60.6%) when removing tracking queries during inference (b). Then we retrain MOTR as a sole detection task by removing tracking queries (c), and the mAP further increases to 66.1% (+5.5%). This means the DETR-style MOT model has a sparking capability of detection but still struggles with the temporal association of varied appearances, which is the crucial factor of MOT.

We also observe excellent detection performance (70.7%) for MOTRv2, which introduces a pre-trained YOLOX detector. Removing tracking queries during inference brings a slight improvement (1.4%) in mAP, which means MOTRv2 has almost addressed the poor detection issue with high-quality detection priors from YOLOX. However, the introduction of YOLOX brings extra computational burden, unfriendly for deployment. In contrast, we aim to endow the end-to-end MOT model with its own powerful detection capability, rather than introducing any extra pretrained detector.

## 2.2 TRACKING AWARE LABEL ASSIGNMENT

Here we revisit the Tracking Aware Label Assignment (TALA) used to train end-to-end Transformers such as MOTR and TrackFormer for MOT. At time $t - 1$, $N$ queries are categorized into two types: $N_T$ tracking queries $Q_t = \{q_t^1, ..., q_t^{N_T}\}$ and $N_D$ detection queries $Q_d = \{q_d^1, ..., q_d^{N_D}\}$, where $N = N_T + N_D$. All the queries will self-attend each other and then cross-attend the image feature tokens via $L$ decoders, and the output embeddings of the $l$-th decoder are denoted as $E^l = \{e_1^l, ..., e_{N_T}^l\}$ and $F^l = \{f_1^l, ..., f_{N_D}^l\}$. At time $t$, there are $M_G$ ground truth boxes. Among them, $M_T$ are previously tracked objects, denoted as $\hat{E} = \{\hat{e}_1, ..., \hat{e}_{M_T}\}$, which are assigned to $N_T$ tracking queries, where $M_T \leq N_T$ as some objects disappear. Formally, $j$-th tracking em-

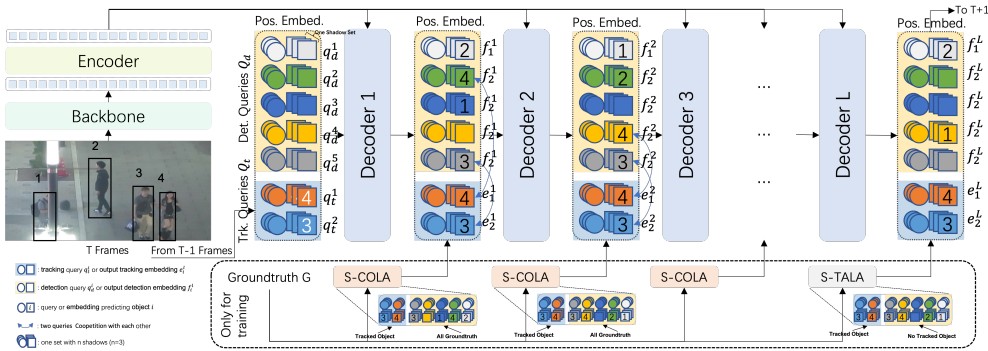

Figure 2: The CO-MOT framework includes a CNN-based backbone network for extracting image features, a deformable encoder for encoding image features, and a deformable decoder that uses self-attention and cross-attention mechanisms to generate output embeddings with bounding box and class information. The queries in the framework use set queries as units, with each set containing multiple shadows that jointly predict the same target. Detection queries and tracking queries are used for detecting new targets and tracking existing ones, respectively. To train CO-MOT, S-COLA and S-TALA are proposed for training only.

bedding $e_j^l$ will be assigned to the same identity as in the previous timestamp if still alive at this moment; otherwise, it will be set to zero (disappearing). Besides, $M_D$ newborn objects, denoted as $\hat{F} = \{\hat{f}_1, ..., \hat{f}_{M_D}\}$, are assigned to $N_D$ detection queries. Specifically, the Hungarian matching algorithm is used to find the optimal pairing between $F^i$ and $\hat{F}$ for each decoder, using a cost function ($L_m = L_f(c) + L_1(b) + L_g(b) \in \mathbb{R}^{N_D \times M_G}$) that takes into account the class scores and box overlapping. Where $L_f(c)$ represents the focal loss for classification, $L_1(b)$ represents the $L_1$ cost of the bounding box, and $L_g(b)$ represents the Generalized Intersection over Union cost.

## 2.3 OVERALL ARCHITECTURE

The entire CO-MOT framework is illustrated in Figure 2. During the forward process, the features of an image in a video are extracted by the backbone and fed into the deformable encoder to aggregate information. Finally, together with the detection and tracking queries, they are used as the inputs of the $L$ layer decoders ($L = 6$ in this paper by default) to detect new targets or track the already tracked targets. It is worth noting that queries contain $(N_T + N_D) \times N_S$ positions ($\mathbb{P} \in \mathbb{R}^4$) and embeddings ($\mathbb{E} \in \mathbb{R}^{256}$) as we use deformable attention. Here $N_S$ is the number of shadow queries for each set, and we will introduce the shadow set concept in the following section. All the queries predict $(N_T + N_D) \times N_S$ target boxes, where $N_S$ queries in a set jointly predict the same target. To train CO-MOT, we employ the COLA and TALA on the different decoders, along with the one-to-set label assignment strategy.

## 2.4 COOPETITION LABEL ASSIGNMENT

Unlike TALA, which only assigns newborn objects to detection queries, we propose a novel COopetition Label Assignment (COLA). Specifically, we assign $M_T$ tracked objects to detection queries in the intermediate decoders, i.e., $l < L$, as illustrated in Figure 2. As shown in the output of the first decoder, the track queries continue to track the 3rd and 4th person. The detection queries not only detect the 1st and 2nd newborns but also detect the 3rd and 4th people. Note that we remain the competition assignment for the $L$-th decoder to avoid trajectory redundancy during inference. Thanks to the self-attention mechanism used between tracking and detection queries, detection queries with the same identity can enhance the representation of the corresponding tracking queries (e.g., grey 3rd helps blue 3rd).

## 2.5 SHADOW SET

In densely crowded scenes, objects can be lost or mistakenly tracked to other objects due to minor bounding box fluctuations. We conjecture that one query for one object is sensitive to prediction

noise. Inspired by previous works such as Group-DETR and H-DETR, we propose the one-to-set label assignment strategy for multi-object tracking, which is significantly different from the one-to-many manner. During tracking, an object is no longer tracked by a single query but by a set of queries, where each member of the set acts as a shadow of each other. Tracking queries are rewritten as $Q_T = \{\{q_T^{1,j}\}_{j=1}^{N_S}, ..., \{q_T^{N_T,j}\}_{j=1}^{N_S}\}$, and detection queries are rewritten as $Q_D = \{\{q_D^{1,j}\}_{j=1}^{N_S}, ..., \{q_D^{N_D,j}\}_{j=1}^{N_S}\}$. The total number of queries is $N \times N_S$. When a particular query in the set tracks the object incorrectly, the other shadows in the same set help it continue tracking the object. In experiments, this strategy prove effective in improving tracking accuracy and reducing tracking failures in dense and complex scenes.

**Initialization.** $P^{i,j} \in \mathbb{R}^4$ and $X^{i,j} \in \mathbb{R}^{256}$, which represent the position and embedding of the $j$-th shadow query in the $i$-th set, are initialized, significantly affecting convergence and final performance. In this paper, we explore three initialization approaches: i) $I_{rand}$: random initialization; ii) $I_{copy}$: initializing all shadows in the same set with one learnable vector, *i.e.*, $P^{i,j} = P^i$ and $X^{i,j} = X^i$, where $P^i$ and $X^i$ are learnable embeddings with random initialization; iii) $I_{noise}$: adding Gaussian noise $\mathcal{N}(0, \sigma_p)$ and $\mathcal{N}(0, \sigma_x)$ to $P^{i,j}$ and $X^{i,j}$, respectively, in the previous approach. In the experiment, we set $\sigma_p$ and $\sigma_x$ to 1e-6. Although the variance between each shadow in the same set is subtle after initialization, it expands to 1e-2 at the end of training. The last approach provides similarity for helping optimization and diversity to improve tracking performance.

**Training.** We propose a shadow-based label assignment method (S-COLA or S-TALA) to ensure that all queries within a set are matched to the same ground truth object. Take S-COLA as an example: we treat the set as a whole and select one query as a representative based on certain criteria to participate in subsequent matching. Specifically, for tracking queries $Q_t$, the tracked target in the previous frame is selected to match with the whole set; For detection queries $Q_d$, we first calculate the cost function ($L_{sm} \in \mathbb{R}^{N_D \times N_S \times M_G}$) of all detection queries with respect to all ground truth. We then select the representative query using a strategy $\lambda$ (*e.g.,* Mean, Min, and Max) for each set, resulting in $L_m = \lambda(L_{sm}) \in \mathbb{R}^{N_D \times M_G}$. $L_m$ is then used as an input for Hungarian matching to obtain the matching results between the sets and newborns. Finally, the other shadows within the same set share the representative's matching result.

**Inference.** We determine whether the $i$-th shadow set tracks an object by the confidence score of the selected representative. Here we adopt a different strategy $\phi$ (*e.g.,* Mean, Min, and Max) for representative sampling. When the score of the representative is higher than a certain threshold $\tau$, we select the box and score predictions of the shadow with the highest score as the tracking outputs and feed the entire set to the next frame for subsequent tracking. Sets that do not capture any object will be discarded.

# 3 EXPERIMENT

## 3.1 DATASETS AND METRICS

**Datasets.** We validate the effectiveness of our approach on different datasets, including DanceTrack, MOT17, and BDD100K.

The DanceTrack dataset is used for multi-object tracking of dancers and provides high-quality annotations of dancer motion trajectories. This dataset is known for its significant challenges, such as fast object motion and similar object appearances.

The BDD100K dataset is a large-scale autonomous driving scene recognition dataset used for scene understanding in autonomous driving systems. This dataset provides multiple object categories, such as cars, pedestrians, etc. It can be used to evaluate our model's performance in multi-object tracking across different object categories. The challenges of this dataset include rapidly changing traffic and road conditions, diverse weather conditions, and lighting changes.

The MOT17 dataset is a commonly used multi-object tracking dataset, with each video containing a large number of objects. The challenges of this dataset include high object density, long occlusions, varied object sizes, dynamic camera poses, and so on. Additionally, this dataset provides various scenes, such as indoor, outdoor, and city centers.

**Metrics.** To evaluate our method, we use the Higher Order Tracking Accuracy (HOTA) metric (et al., 2020), which is a higher-order metric for multi-object tracking. Meantime We analyze the contributions of Detection Accuracy (DetA), Association Accuracy (AssA), Multiple-Object Tracking Accuracy (MOTA), Identity Switches (IDS), and Identity F1 Score (IDF1). For BDD100K, to better evaluate the performance of multi-class and multi-object tracking, we use the Tracking Every Thing Accuracy (TETA) (Li et al., 2022b), Localization Accuracy (LocA), Association Accuracy (AssocA), and Classification Accuracy(ClsA) metrics. The best results of end-to-end methods are marked in bold. Please pay more attention to the metrics with blue.

Table 2: Comparison to existing methods on the DanceTrack test set. "*" and "+" respectively represent the use of DAB-Deformable backbone and joint training with CrowdHuman. For static images in CrowdHuman dataset, we apply random shifts as in CenterTrack to generate video vlips with pseudo tracks.

| | Source | HOTA | DetA | AssA | MOTA | IDF1 |
|---|---|---|---|---|---|---|
| Non-End-to-end | | | | | | |
| CenterTrack (Zhou et al., 2020) | ECCV'20 | 41.8 | 78.1 | 22.6 | 86.8 | 35.7 |
| TransTrack (Sun et al., 2020) | arXiv'20 | 45.5 | 75.9 | 27.5 | 88.4 | 45.2 |
| FairMOT (Zhang et al., 2021) | IJCV'21 | 39.7 | 66.7 | 23.8 | 82.2 | 40.8 |
| QDTrack (Fischer et al., 2022) | CVPR'21 | 54.2 | 80.1 | 36.8 | 87.7 | 50.4 |
| TraDeS (Wu et al., 2021) | CVPR'21 | 43.3 | 74.5 | 25.4 | 86.2 | 41.2 |
| ByteTrack (Zhang et al., 2022b) | ECCV'22 | 47.7 | 71.0 | 32.1 | 89.6 | 53.9 |
| GTR (Zhou et al., 2022) | CVPR'22 | 48.0 | 72.5 | 31.9 | 84.7 | 50.3 |
| MT-IoT$^+$ (Yan et al., 2022) | arXiv'22 | 66.7 | 84.1 | 53.0 | 94.0 | 70.6 |
| OC-SORT (Cao et al., 2023) | CVPR'23 | 55.1 | 80.3 | 38.3 | 92.0 | 54.6 |
| C-BIoU (Yang et al., 2023) | WACV'23 | 60.6 | 81.3 | 45.4 | 91.6 | 61.6 |
| MOTRv2$^+$ (Zhang et al., 2023) | CVPR'23 | 69.9 | 83.0 | 59.0 | 91.9 | 71.7 |
| FineTrack (Ren et al., 2023) | CVPR'23 | 52.7 | 72.4 | 38.5 | 89.9 | 59.8 |
| GHOST (Seidenschwarz et al., 2023) | CVPR'23 | 56.7 | 81.1 | 39.8 | 91.3 | 57.7 |
| Walker (Segu et al., 2024) | ECCV'24 | 52.4 | 36.1 | 76.5 | 89.7 | 55.7 |
| GeneralTrack (Qin et al., 2024) | CVPR'24 | 59.2 | 82.0 | 42.8 | 91.8 | 59.7 |
| MotionTrack (Xiao et al., 2024b) | arXiv'24 | 58.2 | 81.4 | 41.7 | 91.3 | 58.6 |
| ConfTrack (Jung et al., 2024) | WACV'24 | 56.1 | - | - | 89.6 | 56.2 |
| MambaTrack (Xiao et al., 2024a) | arXiv'24 | 56.8 | 80.1 | 39.8 | 90.1 | 57.8 |
| Hybrid-SORT (Yang et al., 2024) | AAAI'24 | 62.2 | - | - | 91.6 | 63.0 |
| UCMCTrack (Yi et al., 2024) | AAAI'24 | 63.6 | - | 51.3 | 88.8 | 65.0 |
| DiffusionTrack (Luo et al., 2024) | AAAI'24 | 52.4 | 82.2 | 33.5 | 89.3 | 47.5 |
| End-to-end | | | | | | |
| MOTR (Zeng et al., 2022) | ECCV'22 | 54.2 | 73.5 | 40.2 | 79.7 | 51.5 |
| DNMOT (Fu et al., 2023) | arXiv'23 | 53.5 | - | - | 89.1 | 49.7 |
| MeMOTR (Gao & Wang, 2023) | ICCV'23 | 63.4 | 77.0 | 52.3 | 85.4 | 65.5 |
| MeMOTR* (Gao & Wang, 2023) | ICCV'23 | 68.5 | 80.5 | 58.4 | 89.9 | 71.2 |
| MOTRv3$^+$ (Yu et al., 2023) | arXiv'23 | 68.3 | - | - | 91.7 | 70.1 |
| SUSHI (Cetintas et al., 2023) | CVPR'23 | 63.3 | 80.1 | 50.1 | 88.7 | 63.4 |
| MambaTrack+ (Huang et al., 2024) | arXiv'24 | 56.1 | 80.8 | 39.0 | 90.3 | 54.9 |
| OuTR (Liu et al., 2024) | arXiv'24 | 54.5 | - | - | 88.3 | 55.7 |
| DiffMOT (Lv et al., 2024) | CVPR'24 | 62.3 | **82.5** | 47.2 | **92.8** | 63.0 |
| ByteSSM (Hu et al., 2024) | arXiv'24 | 57.7 | 81.5 | 41.0 | 92.2 | 57.5 |
| CO-MOT | - | 65.3 | 80.1 | 53.5 | 89.3 | 66.5 |
| CO-MOT$^+$ | - | **69.4** | 82.1 | **58.9** | 91.2 | **71.9** |

## 3.2 IMPLEMENTATION DETAILS

Our proposed label assignment and shadow concept can be applied to any e2e-MOT method. For simplicity, we conduct all experiments on MOTR. It uses ResNet50 as the backbone to extract image features and employs a Deformable encoder and Deformable decoder to aggregate features and predict object boxes and categories. We also use the data augmentation methods employed in MOTR, including randomly clipping and temporally flipping video segments. To sample a video segment for training, we use a fixed sampling length of 5 frames and a sampling interval of 10

frames. The dropout ratio in attention is set to zero. We train all experiments on 8 V100-16G GPUs, with a batch size of 1 per GPU. For DanceTrack and BDD100K, we train the model for 20 epochs with an initial learning rate of 2e-4, reducing the learning rate by a factor of 10 every eight epochs. We use 60 initial queries for a fair comparison with previous work. For MOT17, we train the model for 200 epochs, with the learning rate reduced by a factor of 10 every 80 epochs. We use 300 initial queries due to the large number of targets to be tracked.

### 3.3 COMPARISON WITH STATE-OF-THE-ART METHODS

**DanceTrack.** Our method presents promising results on the DanceTrack test set, as evidenced by Table 2. Without bells and whistles, our method achieves an impressive HOTA score of 65.3%. Compared to other e2e-MOT methods with the ResNet50 backbone, CO-MOT achieves remarkable performance improvements(*e.g.,* 11.1% improvement on HOTA compared to MOTR, 11.8% compared to DNMOT, and 1.9% compared to MeMOTR). Although it falls short of MeMOTR*, it is worth noting that MeMOTR* utilizes the more powerful DAB-Deformable-DETR. In comparison with Non-e2e-MOT methods, our approach demonstrates significant improvements across various tracking metrics. For instance, when compared to the state-of-the-art UCMCTrack, CO-MOT achieves a 1.7% improvement in HOTA and 1.5% improvement in AssA. Our approach can avoid tedious parameter adjustments and ad hoc fusion of two independent detection and tracking modules. It realizes automatic learning of data distribution and global optimization objectives.

With joint training on CrowdHuman dataset, our method CO-MOT$^+$ achieves even higher performance with 69.4% HOTA. This is 1.1% improvement over MOTRv3$^+$ with the ResNet50 backbone. Compared to CO-MOT(65.3% vs 69.4% HOTA), we can conclude that increasing the dataset size can lead to further improvements in tracking performance. Additionally, it performs on par with the state-of-the-art Non-e2e-MOT method MOTRv2$^+$, which incorporates an additional pre-trained YOLOX detector into MOTR.

As shown in Table 1, CO-MOT achieved the mAP of 69.1, significantly higher than MOTR's 42.5, and slightly lower than MOTRv2's 70.7. As research in this field continues, models like CO-MOT will likely play a crucial role in advancing the state-of-the-art in multiple-object tracking, offering more reliable and efficient solutions for a variety of applications.

Table 3: Comparison to existing methods on various datasets.

(a) MOT17 Test Dataset

| | HOTA | AssA | MOTA | IDF1 |
|---|---|---|---|---|
| Non-End-to-end | | | | |
| CenterTrack | 52.2 | 51.0 | 67.8 | 64.7 |
| TransTrack | 54.1 | 47.9 | 74.5 | 63.9 |
| FairMOT | 59.3 | 58.0 | 73.7 | 72.3 |
| QDTrack | 63.5 | 64.5 | 77.5 | 78.7 |
| ByteTrack | 63.1 | 62.0 | 80.3 | 77.3 |
| OC-SORT | 63.2 | 63.2 | 78.0 | 77.5 |
| DiffusionTrack | 60.8 | 58.8 | 77.9 | 73.8 |
| MOTRv2 | 62.0 | 60.6 | 78.6 | 75.0 |
| End-to-end | | | | |
| TrackFormer | - | - | 65.0 | 63.9 |
| MOTR | 57.8 | 55.7 | 73.4 | 68.6 |
| MeMOT | 56.9 | 55.2 | 72.5 | 69.0 |
| MeMOTR | 58.8 | 58.4 | 72.8 | 71.5 |
| DNMOT | 58.0 | - | **75.6** | 68.1 |
| CO-MOT | **60.1** | **60.6** | 72.6 | **72.7** |

(b) BDD100K Validation Set

| | TETA | LocA | AssocA | ClsA |
|---|---|---|---|---|
| Non-End-to-end | | | | |
| DeepSORT | 48.0 | 46.4 | 46.7 | 51.0 |
| QDTrack | 47.8 | 45.8 | 48.5 | 49.2 |
| TETer | 50.8 | 47.2 | 52.9 | 52.4 |
| MOTRv2 | 54.9 | 49.5 | 51.9 | 63.1 |
| End-to-end | | | | |
| MOTR | 50.7 | 35.8 | 51.0 | - |
| CO-MOT | **52.8** | **38.7** | **56.2** | **63.6** |

(c) MOT20 Test Dataset

| | HOTA | AssA | MOTA | IDF1 |
|---|---|---|---|---|
| End-to-end | | | | |
| MeMOT | 54.1 | 55.0 | 63.7 | 66.1 |
| TrackFormer | 54.7 | - | 68.6 | 65.7 |
| CO-MOT | 57.5 | 65.7 | 60.1 | 70.5 |

**BDD100K.** Table 3b shows the results of different tracking methods on the BDD100K validation set. To better evaluate the multi-category tracking performance, we adopt TETA, which combines multiple factors such as localization, association, and classification. Compared with other methods, although the LocA was considerably lower, we achieve superior performance on TETA with an

improvement of 2% (52.8% vs 50.8%), which is benefited from the strong tracking association performance revealed by the AssocA (56.2% vs 52.9%). Compared with MOTRv2, CO-MOT slightly falls behind on TETA, but its AssocA is much better than MOTRv2.

**MOT17.** Table 3a shows the results of the MOT17 test set. Due to the overemphasis on detection performance in MOT17, Non-e2e-MOT methods, starting from ByteTrack, excel at leveraging powerful detectors like YOLOX, achieving excellent detection performance (up to 64.5% DetA) along with other impressive metrics. In this regard, Transformer-based methods, especially e2e-MOT, still have a significant gap in detection performance due to the excessive predictions of dense and small objects in MOT17. On the other hand, e2e-MOT suffers from severe overfitting issues because the MOT17 training set is very small, consisting of only about 5K frames. "Transformers lack some of the inductive biases inherent to CNNs, such as translation equivariance and locality, and therefore do not generalize well when trained on insufficient amounts of data," as mentioned in the ViT paper. MOT17 provides insufficient data to train a Transformer model. Additionally, in MOT17 and DanceTrack, bounding boxes that are less than 0.005 of the image area account for 60.26% and 1.17%, respectively, while bounding boxes that are greater than 0.02 of the image area account for 12.94% and 54.97%, respectively. This highlights that MOT17 primarily comprises smaller targets, which poses a significant challenge for enhancing detection performance with Transformer-based models.

However, despite these challenges, we still achieved a considerable improvement compared to other e2e-MOT methods, reaching a HOTA score of 60.1%. Specifically, we improved the performance of object association, which can be reflected by the AssA and IDF1 metrics. These experimental results further validate the effectiveness of our approach.

**MOT20.** As the End-to-End solution has just emerged in the past year, there are not many methods evaluated on MOT20 that we could find. Here are the ones in Table 3c. Notably, our approach achieves 57.5% HOTA, which is the state-of-the-art in End-to-end tracking methods.

## 3.4 ABLATION STUDY

Table 4: Ablation study on individual CO-MOT components on the DanceTrack Validation Set and MOT17 Test Set. As components are added, the tracking performance improves gradually.

| | COLA | Shadow | DanceTack Validation | | | | | MOT17 Test | | | | |
|---|---|---|---|---|---|---|---|---|---|---|---|---|
| | | | HOTA | DetA | AssA | MOTA | IDF1 | HOTA | DetA | AssA | MOTA | IDF1 |
| (a) | ✗ | ✗ | 56.4 | 71.8 | 44.6 | 79.8 | 57.5 | 57.8 | 60.3 | 55.7 | 73.4 | 68.6 |
| (b) | ✓ | ✗ | 60.2 | 73.2 | 49.7 | 81.8 | 62.4 | 58.5 | 58.0 | 59.2 | 70.3 | 70.7 |
| (c) | ✗ | ✓ | 59.0 | 72.6 | 48.2 | 80.9 | 59.6 | - | - | - | - | - |
| (d) | ✓ | ✓ | 61.8 | 73.5 | 52.2 | 81.7 | 63.3 | 60.1 | 59.5 | 60.6 | 72.6 | 72.7 |

**Component Evaluation of CO-MOT.** Based on the results shown in Table 4, we examine the impact of different components of the CO-MOT framework on tracking performance. Through experimental analysis by combining various components on the DanceTrack (Sun et al., 2022) validation set, we achieve significant improvements over the baseline (61.8% vs 56.4%). By introducing the COLA strategy to the baseline (a), we observe an improvement of 3.8% on HOTA and 5.1% on AssA without any additional computational cost. By incorporating the concept of shadow into the baseline (a), HOTA is improved by 2.6% and AssA is improved by 3.6%. Also, to further illustrate the effectiveness of each component of our method, we also conducte ablation experiments on the MOT17 test dataset, as shown in the Table 4. As components are added, the tracking performance improves gradually.

**COLA.** It is also evident from Table 4 that both COLA and Shadow have minimal impact on DetA (71.8% vs 73.5%), which is detection-related. However, they have a significant impact on AssA (44.6% vs 52.2%) and HOTA (56.4% vs 61.8%), which are more strongly related to tracking. On the surface, our method seems to help detection as it introduces more matching objects for detection, but it actually helps tracking.

To answer this question, we demonstrate the attention weights between detection and tracking queries in Figure 3. The horizontal and vertical axes denote the attention weights after self-attention

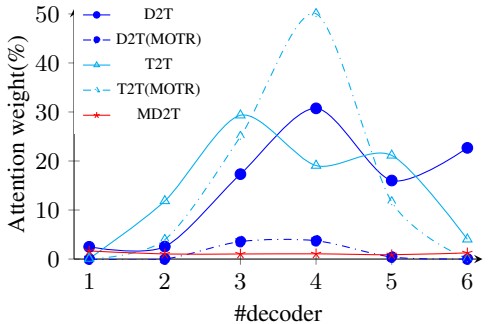

Figure 3: The attention weights between different types of queries on different decoders.

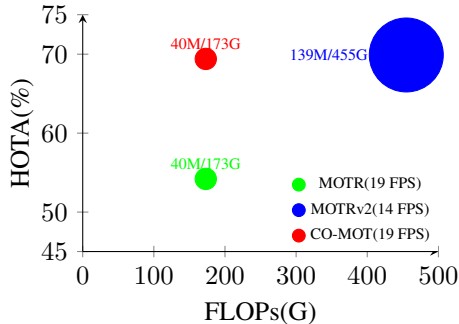

Figure 4: Efficiency comparison for CO-MOT and other end-to-end methods.

between different types of queries on different decoder layers. These weights roughly indicate the contribution of one query to another. In our model, there are a total of 6 decoder layers. T2T represents the contribution of a tracking query to itself. D2T represents the contribution of a detection query predicting the same object to a tracking query. Two bounding boxes with an IOU greater than 0.7 are treated as the same object. MD2T represents the average contribution of all detection queries to a specific tracking query, which serves as a reference metric. D2T(MOTR) and T2T(MOTR) refer to D2T and T2T in the MOTR model. Note that normalized attention weights are with sum of 1.

From Figure 3, it is evident that detection queries make a significant contribution (more than 15%) to their corresponding tracking queries in decoder layers where $L > 2$, even greater than the T2T for #4 and #6 decoders and much higher than the MD2T for all the decoders. This indicates that detection queries pass on the rich semantic information they represent to their corresponding tracking queries, which in turn can be utilized by the tracking queries to improve their tracking accuracy. Compared to CO-MOT, there is almost less information transfer (3.74% vs 30.74%) between detection queries and tracking queries within MOTR. Because in MOTR, detection and tracking queries target different objects, resulting in minimal information exchange, which is straightforward to understand.

Table 5: Performance metrics of COLA inserting different numbers of decoder layers on the DanceTrack validation set.

| $l$ | HOTA | DetA | AssA | MOTA | IDF1 |
|---|---|---|---|---|---|
| $l = 0$ | 59.0 | 72.9 | 48.1 | 81.2 | 59.8 |
| $l = 3$ | 59.6 | 74.3 | 48.0 | 82.8 | 60.5 |
| $l = 5$ | 59.9 | 73.2 | 49.3 | 81.3 | 60.9 |

Table 6: Effect of initialization methods for Shadow queries $I_m$ on the DanceTrack validation set.

| $I_m$ | HOTA | DetA | AssA |
|---|---|---|---|
| $I_m = I_{copy}$ | 60.6 | 73.9 | 50.0 |
| $I_m = I_{noise}$ | 61.5 | 73.1 | 51.9 |
| $I_m = I_{rand}$ | 59.6 | 73.2 | 48.9 |

Furthermore, Table 5 studies the performance impact of COLA inserting different decoder layers on the DanceTrack Validation Set for 5 epochs without Shadow Set. $l = 0$ or $l = 3$ mean that the first layer of the 6-layer decoder or the first three layers use COLA, and the other layers use TALA. It can be seen that deploying COLA in more decoder layers leads to better HOTA.

Table 7: Effect of different $\lambda$ and $\phi$ combinations on the DanceTrack validation set.

| $\lambda$ | | max | | | mean | | | min | |
|---|---|---|---|---|---|---|---|---|---|
| $\phi$ | min | mean | max | min | mean | max | min | mean | max |
| HOTA | 57.6 | 56.4 | 55.1 | 56.7 | 55.2 | 52.0 | 57.5 | 55.9 | 51.5 |
| DetA | 70.7 | 69.3 | 65.4 | 70.6 | 66.5 | 59.0 | 70.8 | 66.4 | 59.3 |
| AssA | 47.3 | 46.1 | 46.7 | 45.9 | 46.1 | 46.1 | 46.8 | 47.2 | 45.0 |

Table 8: Effect of initialization methods for number of Shadows $N_S$ on the DanceTrack validation set.

| $N_S$ | HOTA | DetA | AssA |
|---|---|---|---|
| $N_S = 2$ | 61.5 | 73.1 | 51.9 |
| $N_S = 3$ | 61.8 | 73.5 | 52.2 |
| $N_S = 3$ | 60.8 | 73.8 | 50.3 |

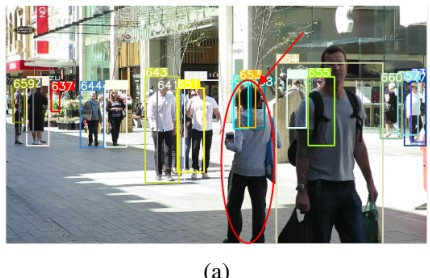
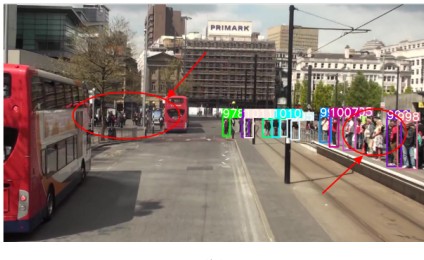

(a)                                                    (b)

Figure 5: Failed cases are often due to the failure to detect the target.

**Shadow Set.** Tables 6, 7 and 8 list ablation experiments related to three hyperparameters of shadow, which are the number of shadows, initialization method of shadows, and representative sampling strategies $\lambda$ and $\phi$. To choose the appropriate option for $\lambda$ and $\phi$, we first set $N_S$ to 5 and train the model only on the DanceTrack training set for 5 epochs using $I_{rand}$ without COLA. Then we try different combinations of $\lambda$ and $\phi$. It can be seen from Table 7 that the combination of $\lambda = max$ and $\phi = min$ yields the best results. That means we use the most challenging query in the set to train the model, leading to discriminative representation learning. To determine the initialization method, we also fix $N_S = 2$ with COLA and find that the best results are achieved using $I_{noise}$. For $I_{rand}$, there is a considerable variation between different shadows within the same set due to random initialization, making convergence difficult and resulting in inferior results. Finally, we try different values of $N_S$ and find that the best results are achieved when $N_S = 3$. When $N_S$ is too large, we observe that convergence becomes more difficult, and the results deteriorate.

## 3.5 EFFICIENCY COMPARISON

In Figure 4, efficiency comparisons on DanceTrack test dataset are made between CO-MOT and MOTR(v2). The horizontal axis represents FLOPs (G) and the vertical axis represents the HOTA metric. The size of the circles represents the number of parameters (M). It can be observed that our model achieves comparable HOTA (69.4% vs 69.9%) with MOTRv2 while maintaining similar FLOPs (173G) and number of parameters(40M) with MOTR. The runtime speed of CO-MOT is much faster (1.4×) than MOTRv2's. Thus, our approach is effective and efficient, which is friendly for deployment as it does not need an extra detector.

## 3.6 LIMITATIONS

Despite the introduction of COLA and Shadow, which improve the tracking effect of MOTR, the inherent data-hungry nature of the Transformer model means that there is not a significant improvement in smaller datasets like MOT17. As shown in Figure 5a, a prominently visible target has not been detected, but this issue has only been observed in the small MOT17 dataset. And due to the scale problem, the detection and tracking performance is poor for small and difficult targets in Figure 5b. In order to further improve the effect, it is necessary to increase the amount of training data or use a more powerful baseline such as DINO.

## 4 CONCLUSION

This paper proposes a method called CO-MOT to boost the performance of end-to-end Transformer-based MOT. We investigate the issues in the existing end-to-end MOT using Transformer and find that the label assignment can not fully explore the detection queries as detection and tracking queries are exclusive to each other. Thus, we introduce a coopetition alternative for training the intermediate decoders. Also, we develop a shadow set as units to augment the queries, mitigating the unbalanced training caused by the one-to-one matching strategy. Experimental results show that CO-MOT achieves significant performance gains on multiple datasets in an efficient manner. We believe that our method as a plugin significantly facilitates the research of end-to-end MOT using Transformer.

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
