# A APPENDIX

## A.1 RELATED WORKS

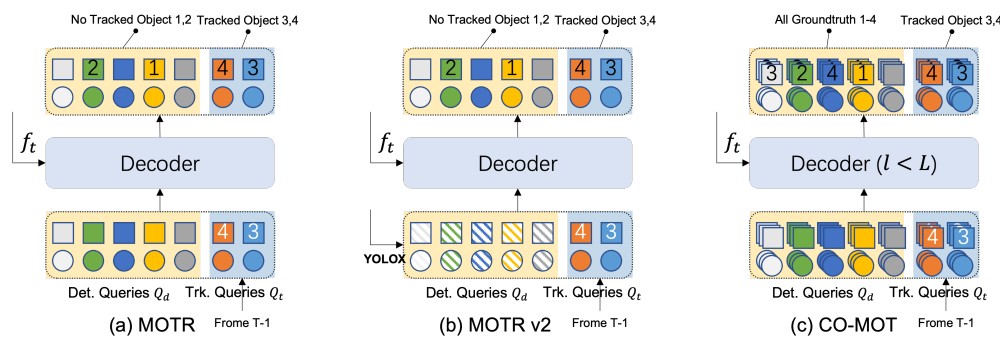

Figure 6: Comparison among MOTR, MOTRv2, and CO-MOT (ours).

**Tracking by detection**: Most tracking algorithms are based on the two-stage pipeline of tracking-by-detection: Firstly, a detection network is used to detect the location of targets, and then an association algorithm is used to link the targets across different frames. However, the performance of this method is greatly dependent on the quality of the detection. SORT (Bewley et al., 2016) is a widely used object tracking algorithm that utilizes a framework based on Kalman Filtering and the Hungarian algorithm; DeepSORT (Wojke et al., 2017) incorporates Reid features extracted by a deep neural network to improve the accuracy and robustness of multi-object tracking based on SORT (Bewley et al., 2016). Afterward, new batch of joint detection and Reid methods are proposed, *e.g.*, JDE (Wang et al., 2020), FairMOT (Zhang et al., 2021); Recently, ByteTrack (Zhang et al., 2022b), OC-SORT (Cao et al., 2023), Strongsort (Du et al., 2023), BoT-SORT are proposed, that have further improved the tracking performance by introducing the strategy of matching with low-confidence detection boxes. While these methods show improved performance, they often require significant parameter tuning and may be sensitive to changes in the data distribution. Additionally, some approaches may require more advanced techniques such as domain adaptation or feature alignment to effectively handle domain shift issues.

**End-to-end tracking**: With the recent success of Transformer in various computer vision tasks, several end-to-end object tracking algorithms using Transformer encoder and decoder modules are proposed, such as MOTR (Zeng et al., 2022) and TrackFormer (Meinhardt et al., 2022). These approaches demonstrate promising results in object tracking by directly learning the associations between object states across time steps. MOTRv2 (Zhang et al., 2023) introduces the use of pre-detected anchor boxes from a YOLOX (Ge et al., 2021) detector to indirectly achieve state-of-the-art performance in multi-object tracking. MeMOTR (Gao & Wang, 2023) introduces a long-term memory that is designed to retain the long-term temporal features for each tracked target. Additionally, it utilizes a temporal interaction module (TIM) to effectively incorporate the temporal information into subsequent tracking processes.

In order to maintain the end-to-end paradigm of MOTR while achieving the performance of MOTRv2, we propose CO-MOT. This method exhibits significant structural differences compared to MOTR (v2), as shown in Figure 6. MOTR(v2) uses TALA supervision for query outputs, meaning the output of detection queries can only match with newborn targets (targets 1 and 2). CO-MOT, on the other hand, uses COLA supervision for $l < L$ decoder layers, allowing the output of detection queries to match not only with newborn targets (targets 1 and 2), but also with already tracked targets (targets 3 and 4). Additionally, CO-MOT employs the Shadow Set method, enabling multiple queries to match the same target. For detailed information, please refer to Figure 2 and Section 2.

**One-to-many label assignment**: DETR (Carion et al., 2020), being a pioneer in employing transformers for computer vision, utilizes a one-to-one label assignment strategy to achieve end-to-end object detection. During training, DETR (Carion et al., 2020) leverages Hungarian matching to compute the global matching cost and thereby assigns each ground-truth box to a unique positive sample. Researchers shifte focus towards enhancing the performance of DETR (Carion et al.,

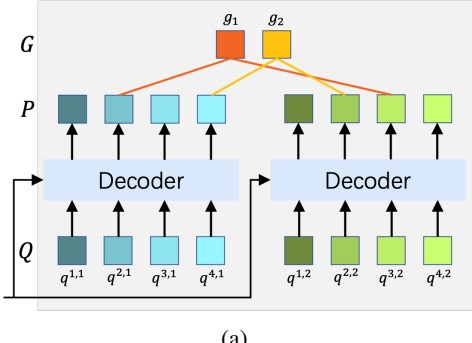 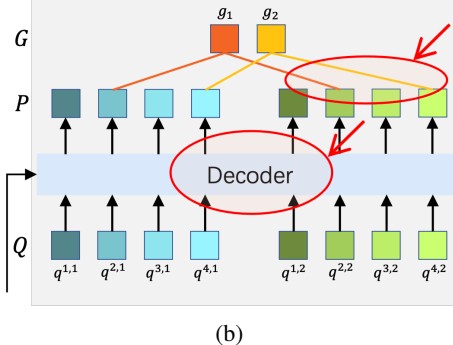

(a)                                              (b)

Figure 7: Compare Group-DETR (a) and Our Shadow Set without tracking queries (b). $Q$: detection queries, where using two groups (sets) of 4 object queries as an example; $G$: ground-truth objects, where two color boxes mean two objects; $P$: predictions. The color lines between $P$ and $G$ correspond to the assignment for ground-truth. The red ellipses and arrows highlight areas of particular interest.

2020), with most efforts concentrated on developing new label assignment techniques. For example, DN-DETR (Li et al., 2022a) building on Deformable DETR, breaks away from the traditional one-to-one matching strategy by introducing noisy ground-truth boxes during training. DINO (Zhang et al., 2022a) builds upon the successes of DN-DETR (Li et al., 2022a) and DAB-DETR (Liu et al., 2022) to achieve an even higher detection performance, putting it at the forefront of current research. Group-DETR (Chen et al., 2023) takes a simpler approach by adopting a group-wise one-to-many label assignment that explores multiple positive object queries. This approach resolves the slow convergence issue often associated with Transformers. Its methodology is similar to the hybrid matching scheme used in H-DETR (Jia et al., 2023). CO-DETR (Zong et al., 2023) introduces multiple additional parallel branches during training to achieve one-to-many allocation. This training scheme helps to overcome the limitations of one-to-one matching and allows for more flexible and accurate object detection in complex real-world scenarios.

It is worth mentioning that Shadow Set and Group-DETR are entirely different concepts. As shown in Figure 7, the first difference is that in Group-DETR, queries from different groups pass through a shared-weight decoder independently. In contrast, Shadow Set queries all pass through the same decoder simultaneously, allowing for self-attention and information exchange among them. The second point, which is also a significant innovation, is that in Group-DETR, the output $P$ from different groups is matched with $G$ using the Hungarian algorithm separately, whereas in Shadow Set, the output $P$ is matched with $G$ using S-COLA without any post-processings. In summary, Group-DETR essentially parallelizes the serial iteration of the decoder; Shadow Set, on the other hand, allocates multiple shadows (which can be understood as multiple similar feature representations) to each target, rather than just a single entity, thereby improving tracking performance.

## A.2 APPLYING COLA AND SHADOW SET TO OTHER E2E-MOT

Table 9: Ablation Experiments of Applying CO-MOT to TrackFormer and MeMOTR on the DanceTrack Validation Set, focusing on the HOTA metric.

| COLA | Shadow | TrackFormer | MeMOTR |
|------|--------|-------------|--------|
| ✗ | ✗ | 41.4 | 51.91 |
| ✓ | ✗ | 47.8(+6.4) | - |
| ✓ | ✓ | 50.7(+9.3) | 53.20(+1.29) |

Table 10: Inference Speeds and Decoder FLOPs for Different Query Configurations. $m * n$ indicates a total of $m$ sets, each containing $n$ shadow set.

| $m * n$ | Inference Speed | Decoder FLOPs |
|---------|-----------------|---------------|
| 60*1 | 91.96 ms | 9.8G |
| 60*3 | 103.11 ms | 10.6G |
| 300*1 | 103.02 ms | 11.6G |

COLA and Shadow Set are model-independent methods that can be applied to any e2e-MOT method, not just MOTR. We incorporate COLA and Shadow Set into the TrackFormer and MeM-

OTR methods, using their respective official default hyperparameters. The results, as shown in Table 9, demonstrate the effectiveness of COLA (+6.4%) and Shadow Set (+9.3%) when applied to TrackFormer. Even with a more powerful backbone (MeMOTR (Gao & Wang, 2023)), there is still 1.29% improvement in HOTA.

## A.3 COST BROUGHT BY SHADOW SETS

While an increase in the number of queries typically raises training and inference costs, the sampling framework used in our paper is similar to that of DETR, consisting of three main modules: ResNet for image feature extraction, the Encoder module for further integration of image features, and the Decoder module for outputting bounding boxes and confidence scores. The increase in queries primarily affects the computation in the decoder layer; however, the decoder contains only six attention layers, which constitute a small portion of the overall model. As shown in the Table 10, the impact on inference speed is negligible (about 6%). The table lists the inference speeds and decoder FLOPs for different query configurations as follows:

## A.4 COMPARISON OF CASES

The Figure 8 illustrates the failure cases of MOTR and denotes which kind of case has been solved by CO-MOT. MOTR has poor detection and tracking performance. First, as shown in Figure 8a, it fails to detect the person in time under the extreme case of bending over. Second, as shown in Figure 8b, due to the tiny visual features when a person stretches out their hand, the detection box is inaccurate, and the model misidentifies it as multiple people. Third, as shown in Figure 8c, the tracking identity switches after the human body is obscured. However, all of the above cases can be solved by CO-MOT, showing the extraordinary performance of MOT.

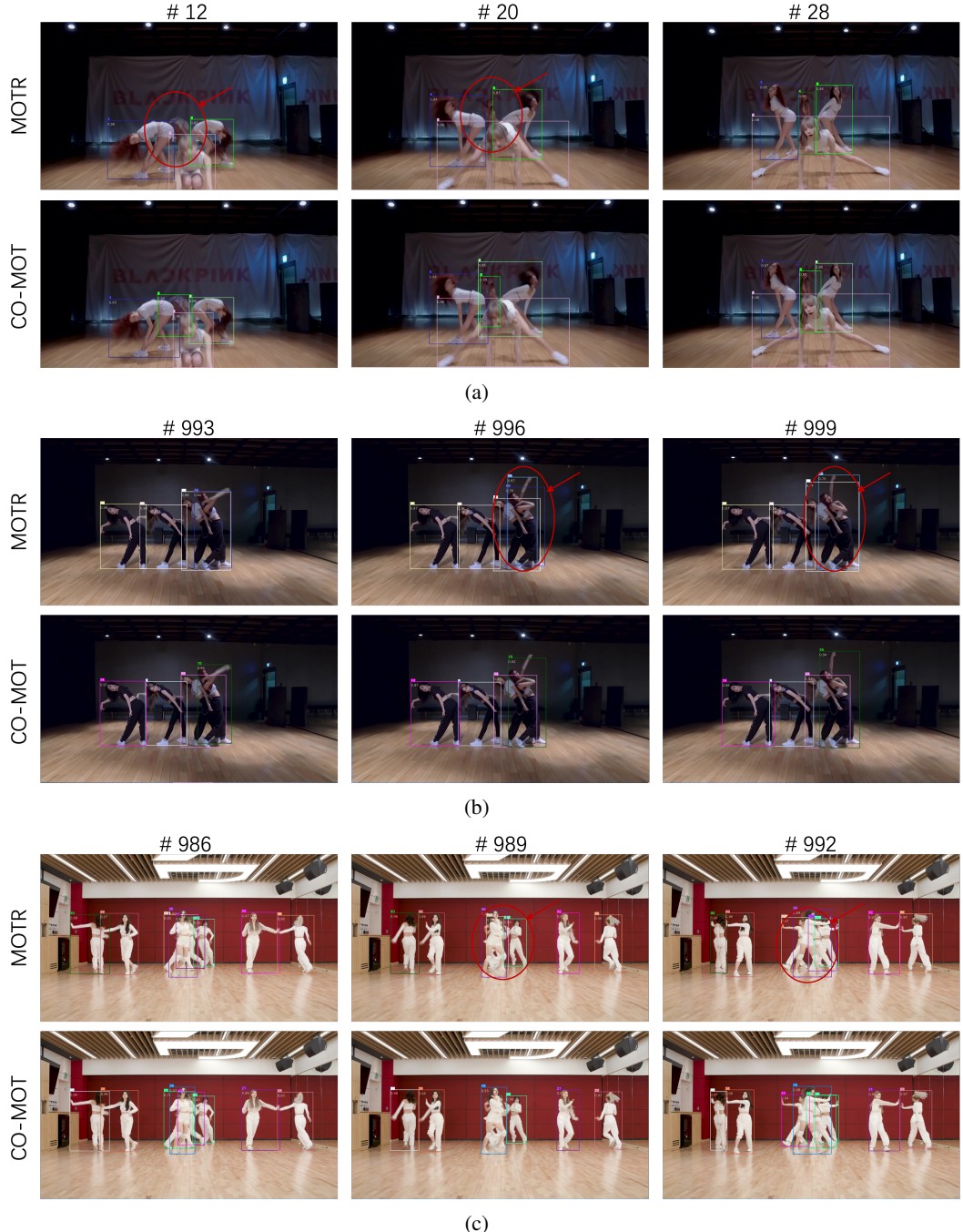

Figure 8: Comparison of Cases: MOTR vs. CO-MOT.