# OpenReview forum: "CO-MOT: Boosting End-to-end Transformer-based Multi-Object Tracking via Coopetition Label Assignment and Shadow Sets"
_ICLR.cc/2025/Conference — ICLR 2025 Poster_

### Official Review · Reviewer_Akzq · 2024-10-27

**Soundness:** 2
**Presentation:** 1
**Contribution:** 2
**Rating:** 6
**Confidence:** 4

**Summary:**

This paper introduces CO-MOT, a novel method aimed at improving end-to-end Transformer-based multi-object tracking (e2e-MOT) through a new coopetition label assignment strategy (COLA) and the introduction of shadow sets. The authors address the issue of unbalanced training in existing e2e-MOT methods, where detection queries often lack positive samples, particularly for newborn objects.

**Strengths:**

- The motivation of the paper is interesting and clearly demonstrated by the experiments.

- The introduction of COLA and shadow sets do mitigate the biased label assignment issue in e2e-MOT. This proposed approach provides a balanced training process, leading to improved performance.

- The authors conduct experiments on three datasets, demonstrating the effectiveness of CO-MOT across different scenarios. The results are robust and consistent, showing improvements over the baseline.

**Weaknesses:**

- The introduction of shadow sets and the coopetition label assignment strategy may increase the computational complexity and training time. The authors should provide a detailed analysis of the computational overhead and discuss potential optimizations. Notably, Fig.4 in the manuscript only presents the Flops, which is not the actual training and inference time. Intuitively, more object queries would bring larger computation costs. Why do shadow sets not?


- Although the proposed method demonstrates strong performance on the tested datasets, it would be advantageous to evaluate CO-MOT on MOT20. The authors assert that the proposed approach enhances detection recall. Thus, the more densely populated nature of MOT20 provides a more suitable context for assessing the effectiveness of the model.

- The authors should investigate the sensitivity of the proposed method to hyperparameters, such as the number of shadow sets and the parameters of the coopetition label assignment strategy. Understanding how these hyperparameters affect performance would provide valuable insights for practical implementation.

- The writing should be improved. For instance, in Fig.3 and Fig.4, the axis titles are overlapped with the axis. The readability of Figures 3 and 4 could be improved by adjusting the axis labels to avoid overlap. This would enhance the overall presentation of the results.

**Questions:**

- Could the authors provide a detailed analysis of the cost brought by shadow sets?
- Could the authors provide the evaluation and discussion on MOT20.

---

> ### Author Response · Authors · 2024-11-20
>
> We sincerely appreciate your time and efforts in reviewing our paper! Based on your review, we added a detailed discussion and additional experiments.
>
> **W1 and Q1:**
> While an increase in the number of queries typically raises training and inference costs, the sampling framework used in our paper is similar to that of DETR, consisting of three main modules: ResNet for image feature extraction, the Encoder module for further integration of image features, and Decoder module for outputting bounding boxes and confidence scores. The increase in queries primarily affects the computation in the decoder layer; however, the decoder contains only six attention layers, which constitute a small portion of the overall model. As shown in the figure, the impact on inference speed is negligible (about 6%). The table lists the inference speeds and decoder FLOPs for different query configurations as follows:
>
> | Query Configuration | Inference Speed | Decoder FLOPs |
> |---------------------|------------------|----------------|
> | 60 *1 | 91.96 ms | 9.8G |
> | 60*3 | 103.11 ms | 10.6G |
> | 300 | 103.02 ms | 11.6G |
> **Table 10:** Inference Speeds and Decoder FLOPs for Different Query Configurations. 60\*1 indicates a total of 60 sets, each containing 1 shadow set. 60\*3 refers to the number of queries used in CO-MOT, while 300 represents the number of queries used in MOTR.
>
> **W2 and Q2:**
> We note that MOT20 indeed has a higher object density, which provides a more challenging environment for evaluating the effectiveness of the model.
>
> This paper primarily focuses on end-to-end tracking methods. Currently, we have only found evaluations of MeMOT and TrackFormer on the MOT20 benchmark, which we have included in Table 3C. At the same time, we conducted a detailed performance analysis on three commonly used benchmarks: DanceTrack, BDD100K, and MOT17. Additionally, Table 1 lists the mAP of various methods, showing that CO-MOT significantly improves the recall of detection boxes.
>
> It is worth mentioning that the phenomena observed in MOT20 and MOT17 are quite similar, with our method outperforming other end-to-end approaches across various metrics. For instance, in the HOTA metric, CO-MOT exceeds MeMOT by 3.4\% and TrackFormer by 2.8\%. These results indicate that CO-MOT demonstrates consistent performance across different datasets.
>
> **W3:**
> We discuss these two questions in Table 9 and Table 5. The optimal number of shadow sets is found to be 3; fewer sets do not contribute effectively, while more sets can introduce negative side effects due to excessive variance within the same set. Additionally, COLA performs best within the l<5 decoders.
>
> **W4:**
> Thank you once again for your valuable feedback. We will make the necessary modifications and adjustments based on your suggestions.
>
> Thank you again for your time and efforts! We show our deepest appreciation for your support of our work. We are always ready to answer your further questions!

---

> > ### Comment · Reviewer_Akzq · 2024-11-25
> > **raising the score**
> >
> > I have reviewed the authors' feedback, and most of the concerns have been adequately addressed. As a result, I am increasing my score to 6. However, I still recommend that the authors improve the quality of their writing and figures.

---

> > > ### Author Response · Authors · 2024-11-26
> > >
> > > Dear Reviewer Akzq,
> > >
> > > We would like to express our profound gratitude for your endorsement of our paper. We will keep polish the paper to meet highest standard. Once again, thank you for your time and effort in reviewing our paper!
> > >
> > > Best, Authors

---

### Official Review · Reviewer_mbwq · 2024-11-03

**Soundness:** 3
**Presentation:** 2
**Contribution:** 2
**Rating:** 6
**Confidence:** 5

**Summary:**

The paper presents an innovative end-to-end Multi-Object Tracking (MOT) framework that aims to enhance transformer-based MOT models. The authors introduce two key contributions: 1. Coopetition Label Assignment (COLA) revises label assignment by allowing detection queries to utilize tracked objects during training in intermediate decoders. This approach boosts feature augmentation for tracking objects with diverse appearances and alleviates the issue of tracking termination. Shadow Set Strategy aims to address training imbalance in one-to-one matching, CO-MOT introduces "shadow sets," which add slight disturbances to each query, thus allowing one-to-set matching. This enhances the discriminative training process and the model's generalization.The proposed method outperforms existing e2e-MOT models on benchmarks like DanceTrack and BDD100K with improved tracking metrics such as HOTA and TETA, demonstrating higher efficiency and inference speed.

**Strengths:**

The paper's strengths lie in its originality, quality, clarity, and significance. It introduces a novel coopetition-based label assignment (COLA) and shadow sets for one-to-set matching, enhancing the robustness of e2e-MOT without requiring additional detectors. The evaluation across multiple datasets, including ablation studies, demonstrates the effectiveness of CO-MOT, establishing its advantages over state-of-the-art models. The paper is well-structured, with clear explanations and helpful visualizations, although further clarification on certain technical aspects could enhance understanding. Overall, CO-MOT significantly improves the efficiency and performance of transformer-based multi-object tracking, making it a valuable contribution to the field.

**Weaknesses:**

While the paper presents valuable contributions, several weaknesses could be addressed to strengthen its impact and clarity:
1. The authors should provide a detailed discussion of the differences between COLA and TALA in Section 2.4, as well as their design in the loss function, to facilitate reader understanding.

2. In the experiments section, the authors need to include comparisons with more methods on the MOT20 and BDD100K datasets.

3. Since the authors analyze the impact of tracking queries on detection performance in transformer-based trackers, if this point serves as one of the motivations, they should compare whether the proposed framework shows improvement in mAP in the experiments.

4. The authors should also analyze the effects of different values of $\lambda$ and $\Phi$ in Section 2.5 on the experimental outcomes.

**Questions:**

I have listed my concerns and questions in Weakness part and hope the response from authors.

---

> ### Author Response · Authors · 2024-11-20
>
> We sincerely appreciate your time and efforts in reviewing our paper! Based on your review, we added a detailed discussion and additional experiments.
>
> **W1: The authors should provide a detailed discussion of the differences between COLA and TALA in Section 2.4, as well as their design in the loss function, to facilitate reader understanding.**
>
> In Figure 6, we briefly explain the differences between COLA and TALA. As shown in Figure 6(a), in COLA, the detection queries can only match with newborns 2 and 1; whereas in TALA, the detection queries can match not only with newborns 2 and 1 but also with tracked individuals 3 and 4. To provide a more detailed description, we will set aside a separate paragraph to discuss the differences between COLA and TALA in depth.
>
> **W2: In the experiments section, the authors need to include comparisons with more methods on the MOT20 and BDD100K datasets.**
>
> In this study, we primarily focus on end-to-end tracking methods and have listed several comparative methods on the DanceTrack and MOT7 datasets.
>
> Due to the relative novelty of our approach, there is currently limited research using MOT20 and BDD100K as evaluation benchmarks. Therefore, we have compiled all known end-to-end tracking methods in the table to provide clear references for readers.
>
> We will continue to monitor developments in this field and consider incorporating more comparisons in future work.
>
> **W3:Since the authors analyze the impact of tracking queries on detection performance in transformer-based trackers, if this point serves as one of the motivations, they should compare whether the proposed framework shows improvement in mAP in the experiments.**
>
> In Table 1, we present the mAP results of representative methods. The mAP of CO-MOT is significantly higher than that of MOTR and slightly lower than that of MOTRv2, indicating that our method is competitive in performance.
>
> **W4: The authors should also analyze the effects of different values of λ and Φ in Section 2.5 on the experimental outcomes.**
>
> We have conducted numerous relevant experiments in Table 8 to explore the effects of different λ and Φ values on the experimental results.
>
> Thank you again for your time and efforts! We show our deepest appreciation for your support of our work. We are always ready to answer your further questions!

---

> > ### Comment · Reviewer_mbwq · 2024-11-25
> >
> > Thanks for the response of author and I'll improve the rating to 6.

---

> > > ### Author Response · Authors · 2024-11-26
> > >
> > > Dear Reviewer mbwq,
> > >
> > > Thank you for your interest in our paper and for your detailed feedback over the past few days!
> > >
> > > we deeply appreciate your efforts in reviewing our work and for the insightful questions that have been invaluable in enhancing our research. We still remain open to addressing any further queries you may have!
> > >
> > > Best regards,
> > >
> > > The Authors

---

### Official Review · Reviewer_d8sy · 2024-11-03

**Soundness:** 3
**Presentation:** 2
**Contribution:** 3
**Rating:** 6
**Confidence:** 4

**Summary:**

This paper addresses the limitations of existing e2e-MOT methods, particularly the unbalanced training issue caused by the label assignment strategy. And it introduces a Coopetition Label Assignment (COLA) strategy and a Shadow Set concept. Through extensive experiments on multiple datasets, it demonstrates superior performance compared to state-of-the-art methods while being more efficient.

**Strengths:**

1. The analysis about the drawbacks of exist e2e-trackers is very interesting, it reveals the negative impact of tracking queries on detection queries.

2. The proposed COLA strategy allows tracked objects to be reassigned to detection queries in decoders, resulting in a significant improvement in tracking performance.

**Weaknesses:**

1. From the public records of OpenReview, it can be seen that this paper was submitted to ICLR2024. The reviewers and AC pointed out many weaknesses last year, while authors have made almost no improvements in the latest version.

2. Many SOTA trackers developed in this year have been overlooked by authors, such as DiffMOT, MambaTrack, TrackSSM, et al. These new methods have made many improvements, and it would be best for the author to provide a comparison with the latest methods.

**Questions:**

See Weaknesses

---

> ### Author Response · Authors · 2024-11-20
>
> We sincerely appreciate your time and efforts in reviewing our paper! Based on your review, we added a detailed discussion and additional experiments.
>
> **W1**:
> We highly value the feedback from the reviewers and AC, and we have made comprehensive improvements in the latest version, including restructuring the paper and adding a significant amount of experimental data, such as more datasets and updated ablation studies.
>
> The ICLR2024 reviewers raised the following issues: see https://openreview.net/forum?id=WLgbjzKJkk. Below, we clarify the points that were raised in previous reviews one by one:
>
> **a: presentation issues, need to refer to baseline MOTR paper.**
> We have significantly reorganized the structure of the entire paper and added relevant citations in the revisions. Additionally, the distance between figures and text has been improved to enhance readability and understanding, for example, by placing Figure 2 close to the corresponding text.
>
> **b: Needs more comparison on BDD100k and DanceTrack.**
> We have added many comparative experiments in the BDD100k and DanceTrack evaluations, as detailed in Tables 2 and 3. These experimental results further validate the effectiveness of our method.
>
> **c: Missing evidence that COLA/Shadows works on other models like Trackformer.**
> We have supplemented the ablation studies on Trackformer, as shown in Table 6. Additionally, COLA and Shadows Set have been introduced into the more powerful backbone, MeMOTR, resulting in performance improvements.
>
> **d: Provide failures cases specific to previous approaches that were solved by CO-MOT.**
>  Figure 8 lists specific cases that CO-MOT has resolved, showcasing the advantages of our method.
>
> **e: Table 2 presentation issues.**
> We have introduced the definitions of non-end-to-end and end-to-end approaches in the introduction for better reader understanding.
>
> **f: Fig 4 show the number of parameters, FLOPS of YOLOX included in MOTR?**
>  Figure 4 displays the number of parameters to provide readers with a clearer understanding of the model's complexity.
>
> **g: Is it necessary to split tracking/detection queries?**
> This issue arises from a misunderstanding of the paper. Tracking queries and detection queries are interdependent in our method and cannot be simply separated.
>
> **h: Missing evaluation on MOT20.**
> We have included the experimental results for MOT20 in Table 3(c).
>
> **i: What if the tracking queries are removed in CO-MOT. Does detection result improve similar to MOTR?**
>  In Table 1(f), we have added the mAP data for CO-MOT, which shows that the detection performance of CO-MOT significantly exceeds that of MOTR.
>
> **j: effect of the number of decoders L on tracking performance.**
> We have added relevant experiments in Table 5 to explore the impact of the number of decoders on tracking performance.
>
> **k: Performance improvement not consistent across different datasets.**
>  Ablation experiments based on multiple baselines across various datasets have been conducted and are presented in Tables 4, 6, and 7, all showing improvements.
>
> **l: Performance worse than MOTRv2 in terms of IDF1 and HOTA.**
>  We provided a detailed explanation in Section 3.4 (COLA) and Figure 3 regarding this phenomenon.
>
> **m: Lack of interpretability of proposed method.**
>  We have included extensive explanations and data analysis in Section 3.3 (MOT7) to enhance the interpretability of our method.
>
> **n: Lack of mathematical formulation for shadow sets -- too many engineering tricks or heuristics.**
>  Detailed explanations and data analysis regarding the design principles of shadow sets are provided in Section 3.4.
>
> **o: missing ablation study on w/ and w/o COLA/shadow set on MOT17 validation set.**
> We have supplemented the relevant experiments in Table 7.
>
> We believe that these revisions have significantly improved the quality and contributions of the paper.
>
>
> **W2:**
> These methods (such as DiffMOT, MambaTrack, TrackSSM, et al) demonstrate excellent performance, indicating that end-to-end tracking approaches are gaining increasing attention. We provide a comparison with our method, CO-MOT, as shown below:
>
> | Method         | HOTA | DetA | AssA | MOTA | IDF1 |
> |----------------|------|------|------|-----|-----|
> | DiffMOT        |  62.3 | **82.5** | 47.2 | **92.8** | 63.0 |
> | MambaTrack     | 55.5 | 80.8 | 38.3 | 90.1 | 53.9 |
> | MambaTrack+    | 56.1 | 80.8 | 39.0 | 90.3 | 54.9 |
> | ByteSSM        | 57.7 | 81.5 | 41.0 | 92.2 | 57.5 |
> | CO-MOT  | **69.4** | 82.1 | **58.9** | 91.2 | **71.9**|
> **Table2 :** Comparison  on the DanceTrack test set.
>
> Our CO-MOT method outperforms the aforementioned methods across multiple key metrics, demonstrating its effectiveness and advantages in the target tracking task. This result further emphasizes the innovation and potential of our approach.
>
> Thank you again for your time and efforts! We show our deepest appreciation for your support of our work. We are always ready to answer your further questions!

---

> ### Comment · Reviewer_d8sy · 2024-11-25
>
> I have carefully read the rebuttal and thank the author for their work. I have compared the differences from last year, and found that the author has made many improvements in both the experiment and writing. While the experimental results are highly impressive on DanceTrack, this work exhibits incremental novelty. I have decided to improve the rating to 6.

---

> > ### Author Response · Authors · 2024-11-26
> >
> > Dear Reviewer d8sy:
> >
> > Thank you for your detailed feedback on our paper and for taking the time to compare our current work with last year's submission. We are very pleased to hear that you recognize the improvements we have made in both the experiments and writing. We also greatly appreciate your acknowledgment of the impressive results on DanceTrack, and we understand your perspective regarding the novelty of our work.
> > We will continue to strive to further enhance the quality of our research.
> >
> > Once again, thank you for your valuable insights and support.
> >
> > Best regards,
> > The Authors

---

### Official Review · Reviewer_rzmb · 2024-11-07

**Soundness:** 3
**Presentation:** 3
**Contribution:** 3
**Rating:** 6
**Confidence:** 5

**Summary:**

The paper tackles end-to-end Transformer-based multiple-object tracking. Previous methods, such as TrackFormer and MOTR, face issues with imbalanced distribution in detection and tracking label assignments, where most objects are assigned to track queries, leaving only a few “newborns” for detection queries. This joint training approach results in weaker detection performance compared to tracking-by-detection methods. To resolve this, the paper proposes a coopetition label assignment strategy to re-balance assignments between track and detection queries. Additionally, it introduces a shadow set that changes the original one-to-one mapping in DETR to a one-to-set mapping, further enhancing tracking performance. Results on various benchmarks demonstrate the effectiveness of this method.​

**Strengths:**

1. The paper is well-written, with a clear and logical structure that makes it very easy to follow.

2. The observation on the disproportional assignment of track and detection queries is insightful, highlighting an important yet often overlooked issue in transformer-based MOT. This analysis provides valuable context for the community.

3. The proposed coopetition label assignment strategy is simple and effective. The paper also demonstrates its effectiveness on multiple Transformer-based MOT frameworks, including TrackFormer and MOTR.

4. The experiments are thorough, covering multiple benchmarks, including more large-scale autonomous driving scenes such as BDD100K, and demonstrating the method’s robustness and practical impact.

**Weaknesses:**

1. To solve the issue of disproportional assignment of track and detection queries, there are also other simpler alternatives. A straightforward option would be to train detection queries jointly on image detection datasets alongside video tracking datasets. For example, detection queries could be trained exclusively on image datasets, treating every object as a new object. An ablation study comparing proposed methods to this simple joint training alternative is appreciated.

2. The paper uses the first 5 decoders to train with all queries, while the last one trains separately on detection and tracking queries. An ablation study could clarify whether a different arrangement, such as using the first decoder for track queries and the last five for all queries, would impact performance. An ablation study regarding this would be helpful for readers to understand the optimal configuration.

3. The applicability of the coopetition label assignment strategy is mostly limited to cases where there is more video data than image data for training, leading to an imbalance in track and detection query assignments. However, in many practical settings, the opposite is true—large-scale [1] and open-vocabulary MOT tasks [2] often have substantially more image detection data than video tracking data. In these cases, common practice in MOT is to use joint training with both image and tracking data, which provides sufficient supervision for detection queries. This is contrary to the paper’s analysis, and it would be beneficial for the authors to also at least discuss these more common scenarios.

[1] Li, Siyuan, et al. "Matching Anything by Segmenting Anything." Proceedings of the IEEE/CVF Conference on Computer Vision and Pattern Recognition. 2024.

[2] Li, Siyuan, et al. "Ovtrack: Open-vocabulary multiple object tracking." Proceedings of the IEEE/CVF conference on computer vision and pattern recognition. 2023.

**Questions:**

The main experiments are still concentrated on small-scale pedestrian tracking datasets. As mentioned on weakness, for other scenarios, we may face different difficulties. Are there any plans to test the model also on large-scale MOT datasets such as TAO [3]?

[3] Dave, Achal, et al. "Tao: A large-scale benchmark for tracking any object." Computer Vision–ECCV 2020: 16th European Conference, Glasgow, UK, August 23–28, 2020, Proceedings, Part V 16. Springer International Publishing, 2020.

---

> ### Author Response · Authors · 2024-11-20
>
> We sincerely appreciate your time and efforts in reviewing our paper! Based on your review, we added a detailed discussion and additional experiments.
>
> **W1: To solve the issue of disproportional assignment of track and detection queries, there are also other simpler alternatives. A straightforward option would be to train detection queries jointly on image detection datasets alongside video tracking datasets. For example, detection queries could be trained exclusively on image datasets, treating every object as a new object. An ablation study comparing proposed methods to this simple joint training alternative is appreciated.**
>
> Yes. For instance, MOTRv2 uses a pre-trained YOLOX to extract detection boxes, significantly enhancing tracking performance. In Table 2, CO-MOT+ uses a combination of Crowdhuman and video data for joint training, which further improves the tracking results on DanceTrack. Therefore, it can be confidently stated that adding a substantial amount of image datasets can indeed enhance tracking performance.
>
> **W2: The paper uses the first 5 decoders to train with all queries, while the last one trains separately on detection and tracking queries. An ablation study could clarify whether a different arrangement, such as using the first decoder for track queries and the last five for all queries, would impact performance. An ablation study regarding this would be helpful for readers to understand the optimal configuration.**
>
> Yes. As shown in Table 5, we have validated the effect of COLA across different decoders. The experiments confirm that using all queries simultaneously on the first five decoders yields the best results, but it is essential to ensure that the last decoder uses the detection and tracking targets obtained from COLA.
>
> **W3: The applicability of the coopetition label assignment strategy is mostly limited to cases where there is more video data than image data for training, leading to an imbalance in track and detection query assignments. However, in many practical settings, the opposite is true—large-scale [1] and open-vocabulary MOT tasks [2] often have substantially more image detection data than video tracking data. In these cases, common practice in MOT is to use joint training with both image and tracking data, which provides sufficient supervision for detection queries. This is contrary to the paper’s analysis, and it would be beneficial for the authors to also at least discuss these more common scenarios.**
>
> Yes. Annotating tracking video data, in particular, requires significant human and financial resources. In contrast, there is currently a wealth of image detection data available, which can be enhanced to further improve tracking performance. Many recent studies have adopted this approach, such as MOTRv2 and MOTRv3, and our CO-MOT+ also further validates this conclusion.
>
> **Q1: The main experiments are still concentrated on small-scale pedestrian tracking datasets. As mentioned on weakness, for other scenarios, we may face different difficulties. Are there any plans to test the model also on large-scale MOT datasets such as TAO [3]?**
>
> Thank you very much for your valuable suggestions. TAO is an excellent benchmark, but it is not well-suited for training models like MOTR and CO-MOT. We have previously conducted experiments on TAO, but the results were not ideal, mainly due to the large amount of unannotated data in TAO, which is more suitable for pre-training or open-vocabulary MOT tasks. However, we will continue to monitor developments in this field and explore more general MOT models.
>
> Thank you again for your time and efforts! We show our deepest appreciation for your support of our work. We are always ready to answer your further questions!

---

> > ### Comment · Reviewer_rzmb · 2024-11-25
> >
> > Thanks for the authors’ feedback. I have read the rebuttal; however, my concerns remain.
> >
> > W1: As the authors acknowledge, training jointly with detection datasets using images yields better performance. As I mentioned, training with image detection datasets also addresses the key issue discussed in the paper: the disproportional assignment of track and detection queries. Hence, I believe it is important to include an ablation study comparing the proposed methods with this simple joint training alternative, as I mentioned earlier.
> >
> > W3 and Q1: The authors mention that their model does not perform well in large-scale scenarios. In such scenarios, we typically have more detection annotations on images than tracking annotations on videos. Therefore, the key issue of the disproportional assignment of track and detection queries seems not to exist. I would expect the authors to provide a discussion on this as a limitation of their method.

---

> > > ### Author Response · Authors · 2024-11-26
> > >
> > > Dear Reviewer rzmb,
> > >
> > > Thank you for your detailed feedback on our paper and for taking the time to review our rebuttal. We greatly appreciate your continued engagement with our work and your insightful comments.
> > >
> > > | Method   | w/ CrowdHuman | COLA | Shadow | HOTA | DetA | AssA | MOTA | IDF1 |
> > > |----------|---------------|------|--------|------|------|------|------|------|
> > > | (a)      |               |      |        | 56.4 | 71.8 | 44.6 | 79.8 | 57.5 |
> > > | (b)      |               | ✅   |        | 60.2 | 73.2 | 49.7 | 81.8 | 62.4 |
> > > | (c)      |               |      | ✅     | 59.0 | 72.6 | 48.2 | 80.9 | 59.6 |
> > > | (d)      |               | ✅   | ✅     | 61.8 | 73.5 | 52.2 | 81.7 | 63.3 |
> > > | (e)      | ✅            |      |        | 56.7 | 73.7 | 43.9 | -    | -    |
> > >
> > > **Table 4:** Ablation study on individual CO-MOT components and the use of CrowdHuman. The baseline used in both MOTRv2 and CO-MOT is the same, which is MOTR.
> > >
> > > **Regarding W1:** We acknowledge the importance of including an ablation study with joint training using detection datasets, and we believe this will provide valuable insights. This experiment has already been conducted in the MOTRv2[1] paper, and we have included the relevant results in row (e). As shown in the table, adding CrowdHuman image data does improve detection performance to some extent, with DetA increasing from 71.8 to 73.7; however, it does not significantly help the tracking-related AssA metric. Through the attention mechanism in our COLA approach, we can transfer the improvement in detection performance to tracking performance, as explained in line 438.
> > >
> > > In summary, simply adding detection data can enhance the model's detection performance to some extent but may not necessarily improve tracking performance. We believe that the COLA strategy proposed in this paper allows for a more significant effect of adding detection data, further enhancing tracking performance.  Tables 4 and 2 demonstrate that **adding detection data and the method proposed in this paper are not mutually exclusive but can complement each other to improve tracking performance.**
> > >
> > > **Regarding W3 and Q1:** Our method can be applied to larger-scale datasets; however, due to the annotation issues present in the TAO dataset (which features many sparse annotations, meaning that some instances of the same category have tracking information while others do not, rather than having dense tracking annotations), it may not be suitable for training standard detection and tracking models. We will clearly state this limitation in our paper and strive to enhance the generalizability of our method in future research.
> > >
> > > Thank you once again for your thorough review and valuable suggestions.
> > >
> > > Best regards,
> > > The Authors
> > >
> > >
> > > [1]  MOTRv2: Bootstrapping end-to-end multi-object tracking by pretrained object detectors. In CVPR, 2023.

---

### Meta-Review · Area_Chair_FVf2 · 2024-12-19

**Metareview:**

This paper presents CO-MOT, a method to enhance end-to-end MOT through a coopetition label assignment strategy (COLA) and shadow sets. Experiments are conducted on multiple datasets, including DanceTrack and BDD100K.

The strengths are: 1) the proposed COLA strategy is simple and effective and 2) new insights on the disproportional assignment of track and detection queries.
The main weaknesses are: 1) missing ablation on comparing with joint training on image detection dataset, and different decoder arrangements, and 2) missing evaluations on scenarios with more image data than video data and comparisons against more recent SOTA trackers.

Since the weaknesses are addressed in the discussion phase and the proposed method is innovative and effective, the AC recommends accepting the paper. In addition, the authors are encouraged to add the ablation studies and additional evaluations to the paper and further polish the writing.

**Additional Comments On Reviewer Discussion:**

The main issues raised by the reviewers include 1) missing ablation on comparing with joint training on image detection dataset and different decoder arrangements, 2) missing evaluations on scenarios with more image data than video data, and comparisons against more recent SOTA trackers.

In the discussion phase, the authors provided detailed discussions and experiments, such as comparisons with more methods, analysis of hyperparameters, and evaluation of additional datasets. All reviewers are satisfied with the feedback and lean toward accepting the paper. The final scores are 6, 6, 6, and 6.

---

### Decision · Program_Chairs · 2025-01-22

Accept (Poster)